# Depth-Aware Adversarial Training for Robust Image Classification

## Abstract

Adversarial examples exploit non-robust, imperceptible features to fool deep neural networks. To explain and address this problem, we propose Depth-Aware Adversarial Training (DAAT), which regularizes model attention to be consistent with scene geometry inferred from monocular depth. Concretely, DAAT leverages a pretrained (frozen) depth estimator to compute depth-gradient maps and imposes an alignment penalty that encourages a Vision Transformer to focus on depth-consistent cues while adversarial examples are generated during training, steering learning away from brittle texture signals toward geometry-aligned evidence. Empirically, on ImageNet-100, On ImageNet-100, DAAT improves $L_\infty$ AutoAttack robust accuracy by 6.96% over standard adversarial training while retaining strong clean performance (80.74%). Theoretically, we further justify DAAT with two analyses: (i) a geometric account showing that small perturbations can distort inferred depth and shift decisions, whereas depth-aligned attention preserves 3D structure in the representation; and (ii) a robust-optimization view in which the alignment term tightens an upper bound on adversarial loss by constraining gradients along depth-inconsistent directions. These results indicate that integrating depth cues into training is a principled route to more robust and interpretable image classifiers, bridging adversarial robustness and 3D vision.

## 1 Introduction

Deep neural networks for image classification are notoriously vulnerable to adversarial perturbations – tiny, human-imperceptible changes to an input image can induce misclassification (Goodfellow et al., 2014; Qian et al., 2022). Adversarial training, which injects adversarial examples during training, remains as one of the most effective defenses (Madry et al., 2017; Zhang et al., 2019; 2021; Rice et al., 2020). However, even adversarially trained models can be brittle and often rely on spurious features that are not human-interpretable (Ilyas et al., 2019). Why are adversarial perturbations so effective? Beyond the well-known hypothesis of linearity in neural networks (Goodfellow et al., 2014), recent studies suggest that robust models tend to align with human perception, emphasizing object shapes and structure over high-frequency textures (Chen et al., 2020; Hoak et al., 2025). This motivates us to explore adversarial robustness from a novel perspective: adversarial attacks may fool models by altering their understanding of 3D structure and object depth in a scene. In human vision, recognizing objects is linked to understanding spatial layout and depth; if adversarial noise confuses a model's depth perception, it might also misidentify objects. We therefore ask whether encouraging a model to focus on proper depth cues can mitigate adversarial vulnerability.

We hypothesize that forcing a model to "see" in terms of depth can improve its adversarial robustness. Adversarial perturbations often work by subtly shifting the input in directions that do not correspond to real changes in the scene's geometry (so-called "non-robust features" (Ilyas et al., 2019)). As a motivating example, Figure 1a shows how an adversarial perturbation applied to an image causes a trained depth model to predict noticeably altered depth values. Even though the perturbation is nearly invisible, the model perceives some regions as significantly closer or farther than in the original image (see red and blue regions in the signed difference map). This suggests that adversarial attacks might deceive vision models by perturbing features related to spatial depth and shape. We also observe that the classifier's attention shifts correspondingly. Figure 1b demonstrates a classifier's Grad-CAM (Selvaraju et al., 2017) visual explanation on a clean vs. adversarial image. The clean image (a man holding a fish) is correctly recognized (class "tench") with a Grad-CAM

highlighting the fish and man. However, after an adversarial perturbation, the model misclassifies the image as a "hen," and the Grad-CAM heatmap drastically changes – the model's focus is misdirected to background foliage and random regions. This implies a misalignment between the model's feature sensitivity and the true 3D structure of objects. If we can realign the model's feature extraction with genuine structure – e.g. object boundaries and surface normals indicated by depth gradients – the model may become less sensitive to superficial pixel-level tweaks. In other words, a model that attends to an object's shape (which correlates with depth discontinuities) should be harder to fool without making a geometrically meaningful change to the image.

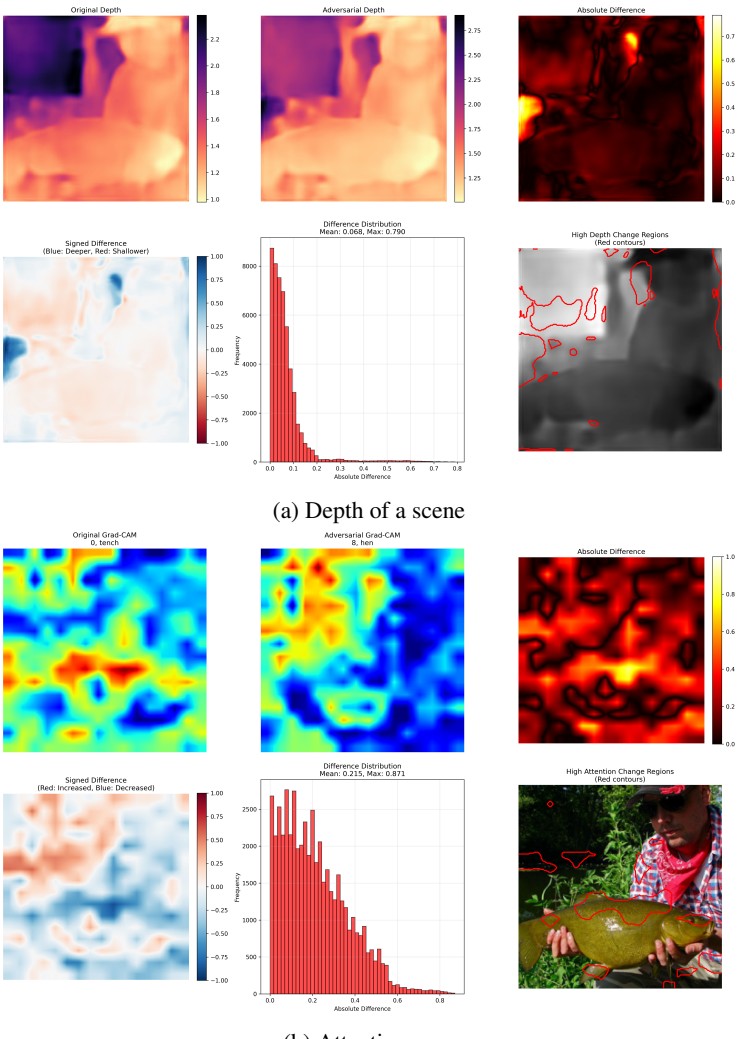

(a) Depth of a scene

(b) Attention map

Figure 1: Illustration of how an adversarial perturbation can alter a model's perceived depth of a scene (a) and attention map (b).

To validate this idea, we develop Depth-Aware Adversarial Training (DAAT). In DAAT, we train a Vision Transformer classifier on adversarial examples generated via PGD, augmented with a depth alignment loss. During training, a fixed pre-trained depth estimation model (DINOv2 + Dense Prediction Transformer (DPT) (Oquab et al., 2023)) computes a depth map for each input; we then compute the spatial gradient of the depth map to identify important 3D edges. Simultaneously, we extract the ViT's attention map (specifically, multi-head self-attention aggregated across layers) for the input. We apply a regularization term that encourages the model's attention to align with the depth edges – effectively penalizing attention on regions that are flat in depth or off the true object contours. By coupling attention with depth cues in the loss, the model learns to emphasize features

coincident with physical object boundaries, which are inherently more robust to perturbations that do not respect those boundaries.

Our contributions are as follows. (1) We propose a novel adversarial defense, DAAT, that integrates monocular depth estimates into the training of an image classifier. To our best knowledge, this is the first attempt to explicitly tie model attention to scene depth for improving robustness in image classification. (2) We conduct an empirical study on ImageNet-100 (a 100-class subset of ImageNet) using a ViT-B/14 model, showing that DAAT significantly improves robust accuracy under strong $L_\infty$ PGD attacks compared to standard adversarial training (PGD-AT). Importantly, this robustness gain comes with minimal loss in clean accuracy (results in Section Experiments). Qualitatively, we observe that DAAT indeed causes the model to focus on semantically meaningful regions (e.g. object silhouettes), and this focus is maintained even under attacks. (3) We provide a theoretical perspective supporting DAAT. Geometrically, aligning attention with depth gradients means the model's decisions depend on stable 3D structures, making it harder for adversarial noise to induce a class change without also causing a noticeable geometric distortion. From a robust optimization perspective, we argue that the depth alignment term acts as a regularizer that limits the model's sensitivity to adversarial perturbations, tightening the known trade-off between robustness and accuracy by biasing the model toward "robust features" (Ilyas et al., 2019). We formalize these intuitions in Section Theoretical Justification.

In summary, DAAT demonstrates that integrating geometry into adversarial training can be a powerful way to bolster model security. We hope our work paves the way for future research on multi-modal and geometry-aware defenses for robust vision.

## 2 RELATED WORK

### 2.1 ADVERSARIAL ATTACKS AND TRAINING

Ever since the discovery of adversarial examples (Szegedy et al., 2013), many attack methods have emerged, from fast one-step methods Goodfellow et al. (2014) to powerful multi-step approaches like Projected Gradient Descent (PGD) and C&W Madry et al. (2017); Carlini & Wagner (2017) and AutoAttack (Croce & Hein, 2020). To defend against these threats, adversarial training has become the de facto standard (Goodfellow et al., 2014; Madry et al., 2017; Zhang et al., 2019; 2021; Peng et al., 2023). In adversarial training, models are trained on adversarially perturbed inputs generated on-the-fly. PGD-AT advocated a min-max optimization framework to withstand strong iterative attacks (Madry et al., 2017). While effective, standard adversarial training often degrades clean accuracy and can overfit to the specific norm-bounded threat model (Rice et al., 2020; Liu et al., 2025). To improve robust generalization, researchers have proposed regularization schemes and objective trade-offs, which adds a term for the gap between natural and adversarial predictions (Zhang et al., 2019). Robust architecture (RA) Peng et al. (2023) offers a comprehensive analysis of how different model structures influence robustness, proposing a robust structure. From the data aspect, diffusion models have been proven to be a good choice to generate training data that improves the robustness of the model Rebuffi et al. (2021); Wang et al. (2023).

### 2.2 ATTENTION ALIGNMENT IN ROBUST MODELS

Several works have explored the idea of aligning internal model representations between clean and adversarial examples to improve robustness. Adversarial Logit Pairing (ALP) encourages the output logits for a natural image and its adversarial counterpart to be similar (Kannan et al., 2018). Beyond logits, attention and feature maps can also be paired. Zagoruyko & Komodakis (2016) introduced attention transfer for model compression, and later AT+ALP was proposed, which explicitly penalizes differences in spatial attention maps between clean and adversarial images (Goodman et al., 2019). Their method improved robust accuracy on small datasets by forcing the model to "look" at the same regions for an image and its perturbed version. Our approach similarly focuses on attention, but instead of matching adversarial to clean attention, we align adversarial attention to an external signal: the depth-based salient regions of the image. This leverages a fixed prior (scene geometry) rather than the model's own clean image features, and is complementary to methods like AT+ALP.

Another related idea is input gradient regularization. Ross & Doshi-Velez (2018) showed that penalizing the magnitude of the gradient of the loss w.r.t. input (or encouraging alignment with human-

attention masks) can both improve interpretability and robustness. Our depth alignment loss can be seen as a specialized form of input-gradient regularization that doesn't just shrink gradients arbitrarily, but rather shapes them to coincide with meaningful depth edges.

## 2.3 Depth and Robust Vision

In human vision, understanding 3D structure is fundamental to recognizing objects; analogously, incorporating geometric information might strengthen machine perception against anomalous inputs. Prior work has utilized depth information mostly in multi-task or multi-modal contexts (e.g. improving object detection in autonomous driving using LiDAR or stereo depth). In the adversarial domain, recent studies on 3D tasks suggest that depth cues can aid robustness. For example, DART3D, a depth-aware adversarial training method for monocular 3D object detection, was proposed to improve robustness by jointly training detection and depth estimation networks (Li et al., 2023). In 2D classification, however, the use of depth for adversarial defense has been unexplored. Our work is the first to inject monocular depth cues into an image classifier's training to enhance adversarial resilience. We note that depth cues have also been linked to shape-based representations. Shi et al. (2020) found that models biased toward shape (as opposed to texture) are significantly more robust to noise and corruptions. Adversarially trained models themselves have been observed to become more shape-biased (Chen et al., 2020). These findings align with our approach: by focusing on depth (which strongly correlates with object shape and boundaries), we inherently increase the model's shape bias and reduce its reliance on brittle texture cues. Our results reinforce this connection between geometric priors and robust feature learning.

## 3 Method: Depth-Aware Adversarial Training (DAAT)

### 3.1 Motivation and Design Principle

Adversarial perturbations often exploit feature directions that do not correspond to meaningful changes in scene geometry. Our central premise is that *geometric structure*—in particular, depth discontinuities that delineate object boundaries and surface transitions—provides a stable, semantically aligned scaffold for recognition. Depth-Aware Adversarial Training (DAAT) instantiates this premise by coupling a classifier's spatial *attention* with a depth-derived *geometric saliency* signal throughout adversarial training. Intuitively, if a model's discriminative evidence is anchored to depth edges, then small pixel-level perturbations that do not induce commensurate geometric change will have limited effect on its decision.

### 3.2 Problem Setup

Let $f_\theta : \mathcal{X} \to \Delta^C$ denote a $C$-class image classifier with parameters $\theta$ and input space $\mathcal{X} \subset \mathbb{R}^{H \times W \times 3}$. We assume access to a fixed depth estimator $g : \mathcal{X} \to \mathbb{R}^{H \times W}$ that maps an image to a scalar depth map (metric or relative). The defense operates under a norm-bounded threat model: for an input $x$ with label $y$, the adversary may choose $\delta$ satisfying $\|\delta\|_p \leq \varepsilon$ to form $x^{\mathrm{adv}} = x + \delta$.

**Attention and Depth-Saliency Fields.** We abstract the model's *spatial attention* as a nonnegative field $A(x) \in \mathbb{R}_{\geq 0}^{H \times W}$ that integrates to one, $\|A(x)\|_1 = 1$, representing the distribution of spatial importance on $x$ (e.g., aggregated self-attention or any differentiable saliency). In parallel, we define a *depth-saliency* field

$$G(x) = \mathcal{N}\big(\mathcal{E}\big(g(x)\big)\big) \in \mathbb{R}_{\geq 0}^{H \times W}, \qquad \|G(x)\|_1 = 1, \tag{1}$$

where $\mathcal{E}$ is a geometry operator that emphasizes depth discontinuities (e.g., a gradient-magnitude or edge functional on the depth map), and $\mathcal{N}$ is a normalization to unit mass. We do not require specific implementations of $A$ or $\mathcal{E}$; DAAT only assumes that $A$ is differentiable w.r.t. $\theta$ and that $G$ is a fixed supervisory signal derived from depth.

### 3.3 Depth–Attention Alignment

DAAT encourages $A(x^{\mathrm{adv}})$ to conform to $G(x^{\mathrm{adv}})$ through a similarity-based penalty. Let $\mathrm{vec}(\cdot)$ flatten a field into a vector and let $\langle \cdot, \cdot \rangle$ denote the Euclidean inner product. We adopt a scale-

invariant cosine objective

$$\mathcal{L}_{\text{align}}(x^{\text{adv}};\theta) \;=\; 1 \;-\; \frac{\big\langle \text{vec}(A(x^{\text{adv}})),\, \text{vec}(G(x^{\text{adv}}))\big\rangle}{\big\|\text{vec}(A(x^{\text{adv}}))\big\|_2 \, \big\|\text{vec}(G(x^{\text{adv}}))\big\|_2}, \tag{2}$$

which attains 0 iff $A$ and $G$ are perfectly aligned (proportional) and grows as they become orthogonal. Other compatible divergences (e.g., Jensen–Shannon, Earth Mover's) can be substituted without altering the training protocol.

### 3.4 EARLY-PHASE REPRESENTATION ALIGNMENT

To facilitate learning of geometry-consistent features early in training, we introduce a transient *feature alignment* term that aligns the student's intermediate representation with that of a frozen, self-supervised teacher (DINOv2). Let $\phi_\theta(x) \in \mathbb{R}^d$ denote the student's representation (e.g., the concatenation of class and patch tokens) and $\phi_T(x) \in \mathbb{R}^d$ the teacher's counterpart. We define the feature alignment loss

$$\mathcal{L}_{\text{feat}}(x;\theta) \;=\; 1 \;-\; \frac{\langle \phi_\theta(x),\, \phi_T(x)\rangle}{\|\phi_\theta(x)\|_2 \, \|\phi_T(x)\|_2}, \tag{3}$$

and activate it only during the initial portion of training. Concretely, for a total of $T$ epochs, we set a time-dependent weight

$$\gamma(t) \;=\; \gamma_0 \, \max\!\Big(0,\, 1 - \tfrac{t}{0.2T}\Big), \qquad t = 0, 1, \ldots, T - 1, \tag{4}$$

which decays linearly from $\gamma_0$ to 0 over the first $20\%$ epochs and remains 0 thereafter. This *bootstrap* encourages the student to inherit a semantics- and shape-aware representation before the adversarial objective dominates, yielding better synergy with depth alignment.

### 3.5 ROBUST OBJECTIVE

DAAT augments the standard min–max formulation of adversarial training Madry et al. (2017) with the alignment term:

$$\min_\theta \; \mathbb{E}_{(x,y)\sim\mathcal{D}}\left[ \max_{\|\delta\|_p \le \varepsilon} \; \underbrace{\mathcal{L}_{\text{ce}}\big(f_\theta(x + \delta), y\big)}_{\text{adversarial classification loss}} \;+\; \lambda \, s(t) \; \underbrace{\mathcal{L}_{\text{align}}(x + \delta;\theta)}_{\text{depth–attention alignment}} \;+\; \gamma(t) \; \underbrace{\mathcal{L}_{\text{feat}}(x;\theta)}_{\text{feature alignment}} \right].$$
$$\tag{5}$$

where $\mathcal{L}_{\text{ce}}$ is the cross-entropy, $\lambda > 0$ weights the alignment, and $s(t) \in [0, 1]$ is a schedule (epoch index $t$) that ramps down the regularizer (e.g., linear warm-start followed by annealing). The depth estimator $g$ is fixed; thus $G(\cdot)$ provides a non-learned geometric prior used only during training. We emphasize that Eq. equation 5 does not prescribe a particular inner maximizer: any norm-bounded adversary (e.g., PGD, CW surrogate) that approximately evaluates the inner maximization is compatible with DAAT.

### 3.6 DESIRABLE PROPERTIES

DAAT confers two complementary robustness properties:

1. **Geometry-consistent sensitivity.** Because $\mathcal{L}_{\text{align}}$ concentrates $A$ on the sparse support of $G$, the input loss gradient $\nabla_x \mathcal{L}$ becomes small away from depth edges. Under norm constraints, the adversary's effective search space shrinks, tightening the worst-case loss bound.

2. **Decision-boundary regularization.** By tying evidence to 3D contours, DAAT biases $f_\theta$ toward decision boundaries that respect object shape. Small appearance changes that do not alter geometry are less likely to cross the boundary, improving perceptual robustness without hand-crafted priors.

Crucially, DAAT remains *model- and attack-agnostic*: it composes with standard architectures (e.g., CNNs, ViTs) and inner maximizers, and requires only a fixed depth oracle during training.

# 4 THEORETICAL ANALYSIS

## 4.1 GEOMETRIC PERSPECTIVE – ALIGNING DECISION BOUNDARIES WITH 3D STRUCTURE

Adversarial perturbations often exploit feature directions that do not correspond to any real semantic or geometric change in the scene (Ilyas et al., 2019). In Figure 1, we saw that a successful perturbation not only fooled the classifier but also altered the depth map in certain regions, effectively simulating a change in the scene's geometry (e.g. parts of the background appeared closer/farther than they were). This observation supports the idea that the classifier's decision boundary in input space was not aligned with the true object boundaries. Ideally, to misclassify an object, one would need to significantly change its shape or depth relationships – something a human would notice. But a non-robust model might make its decision on subtle texture cues; an adversary can tweak those cues without touching the global shape, hence fool the model while the object's depth silhouette remains the same.

DAAT explicitly addresses this by making the model's attention follow the depth gradients. As a result, the model's internal discriminative features are tied to the object's 3D contours. We expect the decision boundary of a DAAT model to cut through input space in a way that respects those contours. An adversary trying to change the model's prediction now has a higher hurdle: they must induce changes to the input that significantly deform the depth map (since only then will the model's attention shift enough to alter the prediction). In effect, DAAT increases the perceptibility of adversarial perturbations – if a perturbation succeeds, it likely also causes a visible geometric distortion, as trivial pixel noise on a flat surface won't suffice. This reasoning parallels findings that adversarial vulnerability is linked to "non-robust features" imperceptible to humans (Ilyas et al., 2019); by forcing the model to use robust, human-aligned features like shape and depth, we remove the model's blind spot that adversaries were exploiting.

More formally, consider two classes whose true separation in an image can be largely characterized by shape (e.g., bird vs airplane may depend on outline). A standard model might latch onto texture (feathers vs metal) which can be altered via pixel-level noise without changing the outline; a depth-aligned model, however, will rely on the outline (depth edges of the object). The adversary's problem becomes one of shape-mimicking – to fool the model, the perturbation must mimic the depth-edge patterns of another class. Such perturbations are necessarily larger in $L_p$ norm and more easily detectable (they resemble adding small structural patterns or "false edges" to the image, as opposed to subtle pixel noise). Thus, the space of effective perturbations is restricted by DAAT. This can be interpreted as the classifier's decision boundary being more aligned with the manifold of real images and with the true geometric differences between classes, making off-manifold moves (which adversarial attacks typically are (Ilyas et al., 2019; Stutz et al., 2019)) less effective.

## 4.2 ROBUST OPTIMIZATION PERSPECTIVE – REGULARIZATION AND LOSS BOUNDS

Adversarial training is often seen as solving a robust optimization: $\min_\theta \mathbb{E}(x, y) \sim \mathcal{D}\left[\max L(f_\theta(x + \delta), y)\right]$. Solving this min-max can be seen as encouraging the loss function to be flat (insensitive) in the neighborhood of each input $x$. One way to achieve a flatter loss landscape is to constrain the gradient $\nabla_x L(f_\theta(x), y)$. Prior work showed that adding a penalty on the input gradient (such as $|\nabla_x L|_2$) can improve robustness by explicitly reducing the worst-case first-order change in loss (Goodman et al., 2019). Our depth alignment loss serves a related purpose: it does not directly minimize the norm of $\nabla_x L$, but it reshapes $\nabla_x L$ to align with $\nabla_x D(x)$ (the depth gradient directions).

To understand the effect, consider a first-order approximation of the adversarial loss increase: $L(f_\theta(x+\delta), y) - L(f_\theta(x), y) \approx \nabla_x L \cdot \delta$. The worst-case $\delta$ under an $L_2$ or $L_\infty$ norm constraint will be aligned with $\nabla_x L$ itself (by Cauchy-Schwarz). Thus $\max_{|\delta| \leq \epsilon} \Delta L \approx |\nabla_x L|*, \epsilon$, where $|\cdot|*$ is the dual norm. If we can enforce that $\nabla_x L$ points along directions of depth edges, we implicitly remove components of $\nabla_x L$ in other directions (e.g. those corresponding to flat regions or high-frequency textures). Depth edges occupy a small fraction of all pixel directions – essentially a sparse mask of salient locations. Aligning $\nabla_x L$ with this sparse support means the gradient is zero in many pixels that lie off any depth edge. Consequently, a perturbation constrained by norm that spreads its budget over many pixels (the most effective way to maximize loss if the gradient were dense) will be much

less effective, because large portions of $\delta$ will be orthogonal to $\nabla_x L$ and contribute nothing to loss increase. In simple terms, the effective dimensionality of the adversary's search space is reduced. The adversary can only significantly increase loss by concentrating $\delta$ on the depth edge regions where $\nabla_x L$ is non-zero. If those regions are limited, the max loss is smaller. This tightens the upper bound on the inner maximization in robust training. By contrast, a standard model might have $\nabla_x L$ spread across many irrelevant pixels (non-robust features), giving the adversary many avenues to exploit.

Another perspective is in terms of regularizing model capacity. The depth alignment loss biases the model to a particular class of functions: those that base their predictions on a limited set of depth-consistent features. This is akin to an inductive bias that can improve generalization. Standard adversarial training alone can sometimes overfit the model to specific perturbation patterns, especially on limited data, resulting in poor robustness on unseen attacks or even on the same attack at test time if the model becomes too specialized. By adding our alignment term, we inject prior knowledge (geometric invariance) that prevents the model from fitting spurious adversarial artifacts. In theory, this should reduce overfitting and improve the model's robust generalization – an intuition supported by our experimental results on validation data.

In summary, from both geometric and optimization standpoints, DAAT works by making the classifier see like a 3D-aware human, focusing on shape and structure. This not only makes attacks more difficult to craft without leaving telltale artifacts, but also simplifies the model's decision logic in a way that improves worst-case loss bounds. A more rigorous theoretical analysis could involve bounding the adversarial risk in terms of alignment loss; we leave a formal proof as future work, but our arguments here outline why depth alignment is a principled regularizer for robustness.

## 5 EXPERIMENTS

### 5.1 SETUP

**Dataset:** We evaluate Depth-Aware Adversarial Training on the ImageNet-100 dataset to validate its effectiveness. ImageNet-100 is a subset of the ImageNet ILSVRC-2012 dataset (Deng et al., 2009) with 100 classes, containing 125K training images and 5K validation images (50 per class). We chose ImageNet-100 to balance computational feasibility and diversity – it is large enough to train a ViT and to include varied natural images, yet smaller than the full 1000-class ImageNet.

**Setup:** Our classifier is a Vision Transformer ViT-Base (patch size 14, 224×224 resolution) Dosovitskiy et al. (2020) initialized from scratch. We train for 300 epochs with AdamW optimizer, base learning rate $5 \times 10^{-4}$, cosine schedule. For adversarial training, we set $\epsilon = 4/255$ ($L_\infty$ norm) and use 10-step PGD with step size $1/255$ for generating adversarial examples online. Following prior work, we use a random start for PGD and the standard cross-entropy loss as the adversarial loss. In DAAT, we add the depth alignment loss with weight $\lambda = 0.1$ (tuned on a small held-out set). We use the DPT-Hybrid depth model from DINOV2 (Oquab et al., 2023) as described earlier, via the official implementation – it was pre-trained on a mixture of datasets and provides robust monocular depth predictions.

**Baselines:** We compare DAAT against Standard Training (no adversarial examples) and PGD Adversarial Training (PGD-AT) (Madry et al., 2017). (our implementation without the depth loss). PGD-AT is the primary baseline to beat.

Table 1: Performance on ImageNet-100

| Method | Clean Acc | PGD-10 | AA Acc |
|---|---|---|---|
| Standard | 81.84 | 10.24 | 8.60 |
| PGD-AT | 78.30 | 52.32 | 46.74 |
| DAAT w/o $\mathcal{L}_{\text{align}}$ | 80.32 | 55.06 | 51.14 |
| DAAT | 80.74 | 56.02 | 53.70 |

**Metrics:** We report Clean Accuracy (on unperturbed validation images) and Robust Accuracy under PGD attacks on the validation set. Specifically, we evaluate with a 10-step PGD at $\epsilon = 4/255$ and

additionally the stronger AutoAttack ensemble (which includes APGD, APGD-T, and gradient-free attacks) to verify robustness. We also examine Grad-CAM visualizations and model attention maps to qualitatively assess where the model "looks" and how it changes under attack.

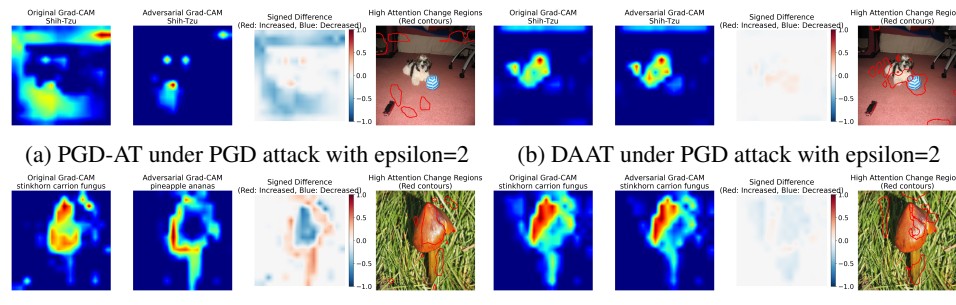

(a) PGD-AT under PGD attack with epsilon=2     (b) DAAT under PGD attack with epsilon=2

(c) PGD-AT under PGD attack with epsilon=4     (d) DAAT under PGD attack with epsilon=4

Figure 2: An illustration of how an adversarial perturbation can alter the attention map of PGD-AT (left) and DAAT (right).

## 5.2 EXPERIMENTAL RESULTS

**Main performance analysis.** Table 1 summarizes the performance. The standard non-robust ViT achieves high clean accuracy (81.84%) but near 10% robust accuracy under PGD and 8.6% under AA, illustrating total vulnerability. With PGD adversarial training, robust accuracy under AA improves to 46.74%, at the cost of some clean accuracy drop to 78.30%. This mirrors observations from prior work: adversarial training produces a much more robust model but slightly less accurate on clean data. DAAT (ours) outperforms plain PGD-AT on robust accuracy under AA attack, achieving 53.70%, a gain of around +6.96% absolute. This indicates that the depth alignment indeed makes the model harder to attack. Notably, DAAT's clean accuracy is 80.74%, which is higher than the PGD-AT baseline – an interesting outcome suggesting that focusing on meaningful depth features can even act as a regularizer improving generalization on natural images. In other words, our model is both more robust and more accurate than the baseline.

**Visual qualitative analysis.** To further illustrate qualitative behavior, Figure 2 compares Grad-CAMs under PGD attacks. With $\epsilon = 2$, both PGD-AT and DAAT keep the label Shih-Tzu, but the PGD-AT Grad-CAM is misaligned with human perception: its high-saliency regions fall largely off the dog and spill into the background (see the signed-difference map and the scattered red contours), whereas DAAT concentrates on the dog's body with only minor shifts. When the budget increases to $\epsilon = 4$, the gap widens: the PGD-AT model flips from stinkhorn, carrion fungus to pineapple, ananas and its attention migrates across unrelated regions, producing strong positive/negative swings and dense red contours over the object. By contrast, DAAT preserves the correct label and keeps its adversarial Grad-CAM close to the clean one. Overall, DAAT stabilizes both attention and predictions, while PGD-AT can focus on non-object regions even at modest $\epsilon$.

**Epoch-wise validation accuracy on ImageNet-100.** We report epoch-wise validation accuracy on ImageNet-100 in Figure 3a and 3b. The left panel reports clean Top-1 accuracy and the right panel reports adversarial Top-1 accuracy (PGD-10 evaluation) on the held-out validation set, measured after each training epoch for three settings: PGD-AT (blue), DAAT without the depth–attention alignment loss $\mathcal{L}_{\text{align}}$ (orange), and full DAAT (green). Both DAAT variants learn faster in early/mid epochs and converge to equal or slightly higher clean accuracy than PGD-AT. On robust accuracy, DAAT consistently outperforms PGD-AT throughout training; the gap opens early and persists at convergence. Adding $\mathcal{L}_{\text{align}}$ provides a small but persistent gain over the ablation, indicating complementary benefits beyond the early DINOv2 feature alignment.

**Robust accuracy vs. attack budget.** As shown in Figure 3c, robust Top-1 (100-step PGD evaluation) decreases monotonically as $\epsilon$ grows, as expected. Across all budgets, both DAAT variants lie above the PGD-AT baseline, with the gap most visible at mid/high budgets ($\epsilon = 4$–10. The full DAAT (green, $\lambda = 0.1$) is consistently competitive with—or slightly better than—the ablation without $\mathcal{L}_{\text{align}}$ (orange), indicating that depth–attention alignment provides small but systematic gains beyond the early DINOv2 feature bootstrap.

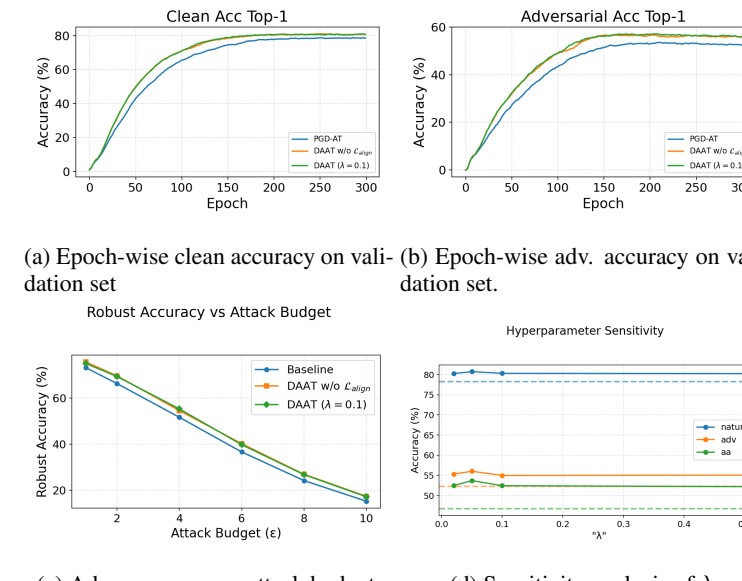

(a) Epoch-wise clean accuracy on validation set

(b) Epoch-wise adv. accuracy on validation set.

(c) Adv. accuracy vs. attack budget.

(d) Sensitivity analysis of $\lambda$.

Figure 3: Validation and robustness analyses. (a) Clean, (b) PGD robust vs. epoch; (c) robust vs. $\epsilon$; (d) sensitivity to $\lambda$.

**Sensitivity to the alignment weight** $\lambda$ Varying $\lambda$ over a wide range yields stable, clean accuracy (blue curve 80%) and robust accuracy that is uniformly above the PGD-AT baselines (dashed lines), as shown in Figure 3d. Robust metrics exhibit a shallow peak around $\lambda \in [0.05, 0.1]$ for both PGD and AutoAttack evaluations, and degrade only mildly at larger $\lambda$. Overall, DAAT's performance is not overly sensitive to the alignment weight: small values already deliver improvements over PGD-AT, while moderate values ($\approx$0.05–0.1) give the best robustness without sacrificing clean accuracy.

## 6 CONCLUSION

We presented Depth-Aware Adversarial Training, a novel defense mechanism that brings depth perception into the loop for robust image classification. DAAT uses a pre-trained depth model to guide the classifier's attention toward meaningful 3D structure, thereby making it harder for adversarial perturbations to divert the model with imperceptible tricks. Through experiments on ImageNet-100 with Vision Transformers, we demonstrated that DAAT substantially improves adversarial robustness over PGD-AT, with improvements on clean accuracy. By aligning model features with physical geometry, the model is more aligned with human visual intuition and less sensitive to high-frequency noise. As a result, the model "looks" at what a human would consider important. We provided theoretical arguments to explain how depth alignment acts as a regularizer that can tighten adversarial loss bounds and force attacks to become more detectable.

This work opens several directions. One is to explore other sources of structural signals – for example, surface normals, segmentation maps, or optical flow – as additional constraints for robust learning. Another direction is to apply DAAT to other architectures (CNNs or larger vision-language models) and other threat models (such as physical world attacks, where depth information could be especially relevant). Moreover, integrating depth cues might improve robustness not only to adversarial noise but also to common corruptions or geometric transformations, a hypothesis to test in future work.

In conclusion, depth – a fundamental aspect of human vision – proves to be a valuable asset in defending against adversarial threats. By training models to respect the depth structure of a scene, we make them more robust, interpretable, and aligned with physical reality. We believe this principle can be extended to build the next generation of trustworthy and resilient computer vision systems.

## ETHICS STATEMENT

Our work does not involve any human subjects, sensitive data, or applications with potential ethical risks. Moreover, this work raises no known ethical concerns.

## REPRODUCIBILITY STATEMENT

To ensure reproducibility, we have provided an anonymized replicate package in the supplementary materials, which contains both the implementation and the train/validation datasets. Details of model ar-chitectures, hyperparameters, and training procedures are described in Section B.1, and all theoretical assumptions and complete proofs are presented in Appendix

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

## A    USE OF LARGE LANGUAGE MODELS (LLMs)

To increase writing clarity and streamline figure narration, we used Large Language Models (LLMs) only for polishing writing and grammar checking. No LLMs were involved in designing experiments, analyzing data, or contributing to the scientific findings of this work.

# B    MORE EXPERIMENTS

## B.1    IMPLEMENTATION DETAILS

In our experiments, we set $\lambda = 0.05$ and use $\epsilon = 4/255$ with 10-step PGD (step size $1/255$) for both training and evaluation (these settings can be adjusted; placeholder) – this is a relatively strong attack on $224 \times 224$ images. After obtaining $x^{\text{adv}}$, we update model parameters $\theta$ by minimizing $\mathcal{L}\text{adv}(x^{\text{adv}}; \theta)$. The depth model $g$ is not updated, and its gradients are not computed (its output is treated as fixed after each forward pass). Note that we compute $G(x^{\text{adv}})$ using the adversarial image itself as input to the depth model. This means the target attention pattern comes from the perturbed image's geometry. An alternative could be to use the original image's depth $G(x)$ as a fixed target; we found using $G(x^{\text{adv}})$ to be effective, presumably because for small $\delta$, $G(x^{\text{adv}})$ remains close to $G(x)$ yet provides some gradient signal when $\delta$ does start to distort depth. In essence, we encourage the model to remain focused on whatever depth edges are present in the current (possibly perturbed) input, instead of drifting to other, less meaningful pixels.

We built on the PyTorch and timm libraries for ViT, and used an open-source DPT depth model pre-trained on a mix of datasets (Oquab et al., 2023). The attention extraction adds some overhead: we hook all 12 layers of ViT-Base, but this is only for computing the regularizer during training. For inference, these hooks are not needed unless one wants to inspect attention. We observed roughly a 2× slow-down in training speed due to depth prediction and attention processing, which is acceptable. The depth model processes 32 images of size 224×224 in about 0.1 seconds on a GPU (batch inference), so it is not a major bottleneck. Memory overhead is also manageable, as the depth model's gradients are not stored. Overall, DAAT training was roughly 2× the time of standard adversarial training in our setup, a reasonable cost given the substantial robustness gains.

## B.2    VISUALIZATION

Figure 4 examines Grad-CAM stability at stronger attacks. (a–b) $\epsilon = 6$, "wing" sample. Both PGD-AT and DAAT flip to Saluki gazelle hound, but their attention behaviors diverge. PGD-AT's heatmap drifts off the airplane and spreads across background regions; the signed-difference map shows large swings and the red contours (high attention-change regions) are widespread. DAAT, while misclassifying, keeps most saliency near the wing and along geometric edges, with smaller signed changes and fewer, more localized contours. Even in failure, DAAT remains more object-centric. (c–d) $\epsilon = 8$, "hare" sample. Both methods retain the correct label (hare). PGD-AT, however, undergoes a broad redistribution of attention over the grass, producing dense contours and strong positive/negative shifts. DAAT's adversarial Grad-CAM stays close to the clean one, with limited changes concentrated around the animal. Takeaway. As the attack budget grows, DAAT consistently suppresses attention drift and localizes inevitable changes to geometry-related regions. When errors occur, they are more interpretable (attention remains on the object), aligning with the depth-anchoring principle behind DAAT.

# C    SIMPLIFIED ANALYSIS UNDER LINEAR AND LOCAL ASSUMPTIONS

To ground these ideas in a more formal analysis, we consider a simplified setting with the following assumptions and notation:

1. Linearized Classifier: Assume $f(x)$ is (locally) linear around a data point $x_0$. For small perturbations, we can write $f(x_0 + \delta) \approx f(x_0) + J(x_0), \delta$, where $J = \nabla_x f$ is the Jacobian (for classification, think of $f$ as the logit score for the true class, or a linear binary classifier for simplicity). Likewise, the loss can be linearized as $\ell(f(x_0 + \delta), y) \approx \ell(f(x_0), y) + \nabla_x \ell^\top \delta$ to first order.

2. Patch-Based Attention: Partition the image into a set of patches or regions indexed by $i \in 1, \ldots, m$ (for example, non-overlapping blocks, or individual pixels as extreme cases). Let $a_i$ denote the attention weight that the model assigns to patch $i$ (how important patch $i$ is for the prediction), and let $g_i$ denote the depth edge strength in patch $i$ (e.g. the average magnitude of $\nabla D$ in that patch, as computed by a Sobel filter). Both $a_i$ and $g_i$ are non-negative and we can normalize $\sum_i a_i = \sum_i g_i = 1$ for convenience.

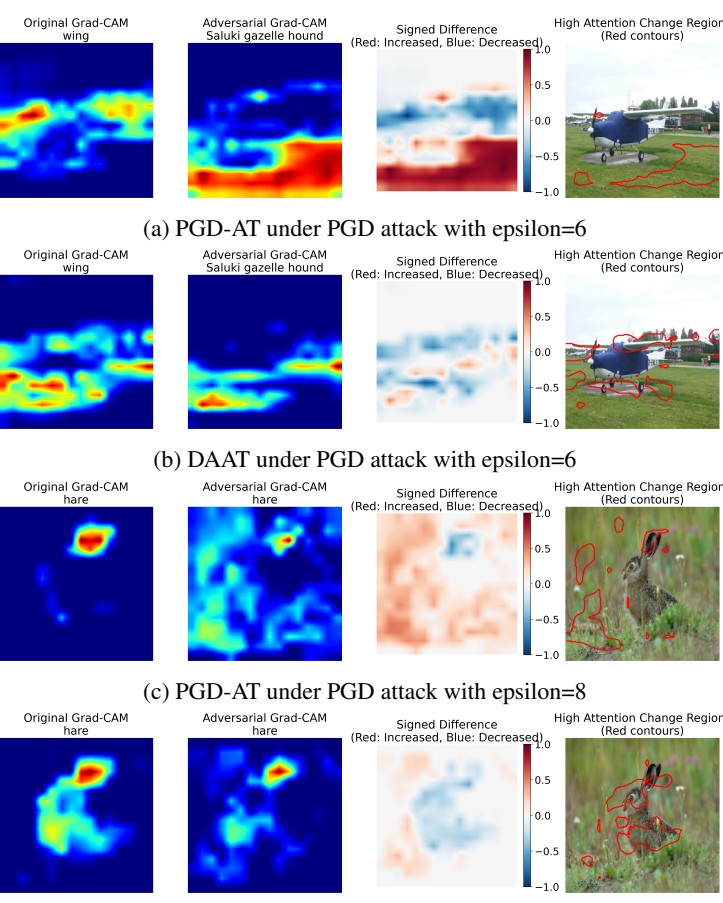

Figure 4: The illustration of how an adversarial perturbation can alter the attention map of PGD-AT and DAAT.

3. Depth-Alignment Loss (simplified): Define $\mathcal{L}_{align} = \sum_i |, a_i - g_i, |$, which measures the deviation between the attention distribution and the depth-gradient distribution. The model is penalized if it places attention on patches that do not have depth edges (or ignores patches that do have edges).

4. Classification on Edges vs Surfaces: Suppose the true task can be largely determined by edge information. For instance, consider that each class differs in the shape/outline of an object, not in the flat texture inside regions. An ideal depth-aligned classifier would then base its decision on differences in patches with high $g_i$ (edges), whereas a non-aligned classifier might also use subtle cues in low-$g$ patches (surfaces) – e.g., texture or noise – which are more vulnerable to perturbation.

Under this setup, we can sketch a theoretical result:

**Proposition 1** *Robustness with Sparse Edge-Focused Features: Consider a linear classifier $f(x) = w^\top x$ and two scenarios for the weight vector $w$: (A) $w$ is aligned with depth edges, meaning $w_i \neq 0$ only for features $i$ that lie on high-depth-gradient patches (and $w_i = 0$ for flat patches); (B) $w$ is unconstrained. Assume the classifier achieves the same margin $\gamma = w^\top x - w^\top x_{boundary}$ on a given example $x$ (where $x_{boundary}$ is the nearest point on the decision boundary along the manifold) in both scenarios. Then, under an $L_\infty$ adversarial perturbation of size $\epsilon$, the worst-case change in the classifier's logit in case (A) is $\Delta_A = \epsilon \sum_{i \in edges} |w_i|$, whereas in case (B) it can be as high as $\Delta_B = \epsilon \sum_i |w_i|$. Because in scenario (A) $w$ has support only on a (typically small) subset of features (edges), $\sum_{i \in edges} |w_i|$ is significantly less than $\sum_i |w_i|$ for an equally expressive classifier in scenario (B). Thus, $\Delta_A \ll \Delta_B$. In other words, an edge-focused classifier offers a tighter upper bound on adversarial logit change.*

*Proof Sketch*: For an $L_\infty$ attack, the maximizing perturbation sets $\delta_i = -\epsilon, \text{sign}(w_i)$ for each feature, attaining $\Delta = \epsilon \sum_i |w_i|$ change in the linear score (this follows from the definition of dual norm: $|\nabla_x f|1$ governs the $L\infty$ vulnerability) arxiv.org . In scenario (A), if $w_i = 0$ for all non-edge features, then $|\nabla_x f|1 = \sum i \in \text{edges}|w_i|$ – the classifier is blind to perturbations on flat regions. In scenario (B), $w$ might be distributed across many features including low-depth-gradient areas, yielding a larger $\ell_1$ norm for the same margin $\gamma$. (Intuitively, to achieve a given classification margin, using a broad set of weak features requires more total weight than using a focused set of strong features on the most informative regions. Texture-based classifiers often sum many small pieces of evidence across the image, accumulating a large $\ell_1$ norm even if the $\ell_2$ norm of $w$ is fixed.) Therefore, $\Delta_B = \epsilon \sum_i |w_i|$ will exceed $\Delta_A = \epsilon \sum_{i \in \text{edges}} |w_i|$. This demonstrates that restricting $w$ to align with depth edges (sparsifying the relevant features) inherently reduces the worst-case impact of an $L_\infty$ perturbation of size $\epsilon$.

The above proposition is a stylized result, but it captures the essence of how DAAT confers robustness. By focusing model capacity on a limited set of geometrically meaningful features, the model's sensitivity to input perturbations is reduced in all other (less meaningful) directions. We saw this through the $\ell_1$ norm of the gradient: depth alignment effectively acts like a regularizer that shrinks the gradient components in non-edge regions to zero. Another way to phrase this is that DAAT introduces an implicit prior that "the classification decision should not drastically change under small input changes unless those changes correspond to real object boundary changes." This prior makes the loss function locally flatter in most directions, except along those that change true object structure.

One can also appeal to the concept of loss landscape stability. A recent theory of adversarial robustness frames it as requiring the loss to be stable (not vary too much) in a neighborhood around natural examples (He et al., 2023). DAAT's alignment term contributes exactly to such stability by eliminating loss spikes due to irrelevant input variation. The auxiliary depth loss acts much like a ridge that pulls the model's decision boundary into alignment with stable, high-level structures. It is a data-dependent regularization: unlike generic smoothness regularizers (which might, say, penalize the norm of $\nabla_x f$ uniformly), the depth alignment specifically targets the most semantically relevant directions for allowed variation (flat regions can vary without affecting class, edges are where variation matters). This yields a more tractable robust optimization problem – essentially narrowing the worst-case loss because the model does not "care" about many of the adversary's available directions.

