# OpenReview forum: "Depth-Aware Adversarial Training for Robust Image Classification"
_ICLR.cc/2026/Conference — ICLR 2026 Conference Withdrawn Submission_

### Official Review · Reviewer_6rZR · 2025-10-20

**Soundness:** 2
**Presentation:** 2
**Contribution:** 2
**Rating:** 2
**Confidence:** 3

**Summary:**

The paper proposes Depth-Aware Adversarial Training (DAAT): during PGD-based adversarial training of a ViT, it aligns the model’s spatial attention with depth-gradient maps from a frozen monocular depth estimator, and briefly bootstraps features via a frozen DINOv2 teacher. The authors motivate DAAT with geometric and robust-optimization analyses arguing that depth-aligned attention concentrates sensitivity on object-boundary structure and shrinks the adversary’s effective search space. On ImageNet-100 with ViT-B/14, DAAT improves AutoAttack robust accuracy by +6.96% over PGD-AT while keeping strong clean accuracy.

**Strengths:**

- Introduces a simple, modular way to inject 3D cues into 2D adversarial training by aligning attention with depth edges (distinct from prior attention/logit pairing that aligns to a model’s own clean saliency). Claims to be the first to tie classifier attention to monocular depth for robustness in image classification.
- Demonstrates meaningful, clean-accuracy-preserving robustness gains on a nontrivial subset of ImageNet, suggesting a practical route to shape/geometry-biased classifiers.

**Weaknesses:**

The most ciritical weakness lies in ignorance of the important previous works. The paper claims novelty as
- *"the first attempt to explicitly tie model attention to scene depth for improving robustness."*,
- *"In 2D classification, however, the use of depth for adversarial defense has been unexplored."*,
- and, *"Our work is the first to inject monocular depth cues into an image classifier’s training to enhance adversarial resilience."*

While the *mechanism* (attention–depth alignment) is distinctive, the high-level idea (injecting auxiliary supervisory signals during adversarial training to improve robustness) sits squarely in a line of work the paper does not engage with experimentally:
- * **Mao et al. (ECCV’20)** showed theoretically and empirically that making models solve *more tasks* can increase robustness (gradient components partially cancel. Robustness improves under multi-task attacks and even single-task attacks).
- * **Ghamizi et al. (AAAI’22)** systematically revisited that claim and found **adding auxiliary tasks is not a guarantee**. Robustness gains depend on task vulnerability/weights, and simple weighting tricks fail against adaptive attacks.
- * **GAT (ICML’23)** explicitly **guides adversarial training with auxiliary tasks** and **regularizes gradient curvature**, solving a **multi-objective** AT problem with MGDA to reach Pareto-optimal trade-offs.

DAAT’s “depth prior” may be viewed as a *particular auxiliary signal*. Without comparing against **(i)** multi-task baselines (e.g., adding a depth-prediction head and training jointly), **(ii)** *guided* AT like **GAT**, or **(iii)** other alignment priors (edges, saliency, normals), it is hard to separate the benefit of **“auxiliary guidance”** from the benefit of **“depth-specific guidance.”** This undercuts the paper’s significance claim and makes it difficult to judge whether DAAT is *principled beyond this one prior*. (The paper itself positions DAAT as first to tie attention to depth and first to inject depth into 2D classification defense, which further heightens the need for comparative evidence.)

- Mao, Chengzhi, et al. "Multitask learning strengthens adversarial robustness." European Conference on Computer Vision. Cham: Springer International Publishing, 2020.
- Ghamizi, Salah, et al. "Adversarial robustness in multi-task learning: Promises and illusions." Proceedings of the AAAI Conference on Artificial Intelligence. Vol. 36. No. 1. 2022.
- Ghamizi, Salah, et al. "Gat: guided adversarial training with pareto-optimal auxiliary tasks." International Conference on Machine Learning. PMLR, 2023.

**Questions:**

I leave some suggestions to deal with the issues I raised.

1. **Multi-task “Depth as Auxiliary” baseline.**
   Attach a lightweight depth-prediction head and train it jointly with classification under AT (Mao’20 setting). Report the same metrics. This isolates whether *attention-alignment* is necessary vs. the simpler *auxiliary-task* route. Cite/ground claims relative to Mao’s positive findings and Ghamizi’s caveats.

2. **GAT baseline.**
   Run **Guided Adversarial Training** on your setup (ImageNet-100 + ViT) using a self-supervised auxiliary (e.g., rotation) and/or a geometry task. Include the **curvature regularizer** and **MGDA weighting**. If DAAT still wins, you can argue the *specific* benefit of aligning to 3D geometry, not just adding an auxiliary.

3. **Ablate the prior:** replace depth edges with *generic edges* (Sobel/Canny), *surface normals*, or *unsupervised saliency* to show whether **3D geometry** specifically drives the gain (not merely “an extra mask to align with”). (Tie back to your own formulation where (G(x)) is any edge functional (E(\cdot)).)

4. **Add Discussion** about the suggested previous works and difference/relationship between your work and each of them.

---

### Official Review · Reviewer_KUYE · 2025-10-23

**Soundness:** 2
**Presentation:** 2
**Contribution:** 2
**Rating:** 2
**Confidence:** 4

**Summary:**

The paper introduces Depth-Aware Adversarial Training (DAAT), which enhances model robustness by aligning attention maps with depth cues from a pre-trained monocular depth estimator. During adversarial training, a depth-attention alignment loss encourages the model to focus on geometrically meaningful regions. Experiments on ImageNet-100 with ViT-B/14 show improved adversarial robustness over standard PGD-based training.

**Strengths:**

The idea of incorporating depth cues is straightforward, and it can be easily integrated into many existing methods to enhance model robustness.

**Weaknesses:**

1. Heavy dependence on a frozen, pre-trained depth estimator. DAAT’s alignment signal requires a depth estimator that is used as a fixed oracle during training. The paper treats the depth as “given” and does not examine how sensitive results are to the depth model’s quality, domain of pretraining, or failure cases. If the depth model gives incorrect edges or systematic biases, DAAT will push attention toward incorrect regions, potentially harm the resulting performance.

2. Evaluation is small-scale and unconvincing (ImageNet-100 only). This dataset is much smaller and less realistic than ImageNet-1k or standard robustness benchmarks. Notably, ImageNet-100 is relatively clean and easy. The leap from ImageNet-100 to large, multi-domain datasets is non-trivial, and the model’s performance in more complex environments remains unclear.

3. Limited baselines and missing comparisons to state-of-the-art robust training methods. The experiments compare only to Standard (no AT) and vanilla PGD-AT, while many stronger baselines and techniques exist. Notably, the paper includes only two references from 2024 and 2025. A more comprehensive review of recent literature is necessary, in addition to addressing the missing comparisons.

4. Missing analysis of failure modes and domain shift. No experiments showing what happens when depth predictions are poor (e.g., textures, synthetic images, artwork, occluded objects) or when the depth model is trained on data that does not match the classifier’s training domain. DAAT could hurt performance under such conditions.

**Questions:**

Please address my comments in paper weaknesses.

---

### Official Review · Reviewer_BmB5 · 2025-11-01

**Soundness:** 3
**Presentation:** 2
**Contribution:** 3
**Rating:** 4
**Confidence:** 5

**Summary:**

This paper proposes a novel defense method called DAAT, which integrates monocular depth estimation into adversarial training to enhance robustness in image classification. A pretrained depth estimator provides depth-gradient maps that guide the Vision Transformer’s attention toward geometry-consistent regions. Extensive experiments demonstrate the effectiveness of the proposed method. The authors also offer geometric and optimization-based theoretical analyses, supported by Grad-CAM visualizations showing stable and semantically aligned attention under attacks.

**Strengths:**

1. The paper introduces a geometry-aware perspective to adversarial robustness by integrating scene depth estimation into training, which is original and appealing.
2. The proposed method only requires a frozen depth estimator and a cosine alignment regularizer, which can be combined with standard PGD-AT pipelines.
3. The geometric analysis explains how DAAT enforces decision boundaries aligned with 3D structure.

**Weaknesses:**

1. Experiments are only on ImageNet-100 with ViT-B/14 and compared to PGD-AT. Evaluations under other attacks, such as FGSM, C&W [1] and black-box attack methods, are not included. In addition, stronger baselines such as TRADES [2], MART [3] should be compared.
2. The computational cost of DAAT is double compared to PGD-AT, which raises concerns for real-world deployment. It is unclear whether DAAT works under fast adversarial training settings (e.g., FGSM-based methods [4]).
3. The method assumes accurate depth predictions, but real-world images may yield noisy or biased depth maps. The impact of using different estimators is not discussed.
4. Eq. (5) defines three losses, classification, depth attention alignment, and feature alignment, but the authors do not explain how to balance or schedule them. The sensitivity to $\lambda$ and $\gamma$ is underexplored.

[1] Carlini N, Wagner D. Towards evaluating the robustness of neural networks[C]//2017 ieee symposium on security and privacy (sp). Ieee, 2017: 39-57.

[2] Zhang H, Yu Y, Jiao J, et al. Theoretically principled trade-off between robustness and accuracy[C]//International conference on machine learning. PMLR, 2019: 7472-7482.

[3] Wang Y, Zou D, Yi J, et al. Improving adversarial robustness requires revisiting misclassified examples[C]//International conference on learning representations. 2019.

[4] Zhang Y, Zhang G, Khanduri P, et al. Revisiting and advancing fast adversarial training through the lens of bi-level optimization[C]//International Conference on Machine Learning. PMLR, 2022: 26693-26712.

**Questions:**

See Weaknesses.

---

### Official Review · Reviewer_F6kN · 2025-11-03

**Soundness:** 1
**Presentation:** 3
**Contribution:** 2
**Rating:** 2
**Confidence:** 3

**Summary:**

This paper proposes a novel adversarial training method that leverages depth information from the input image. In particular, the proposed method introduces additional loss terms to align the model’s spatial attention (either from the model or a specific attribution method) with the depth saliency computed from the depth estimator. The authors provide additional discussion to justify the effectiveness of the proposed idea. Through an experiment, the paper demonstrates the improvement from the proposed method. Further experiments, such as qualitative analysis and robust accuracy over varying attack strengths, are also presented.

**Strengths:**

1. To the best of my knowledge, the proposed method is novel.
2. The main idea to align the model’s attention (to other sources of shape information) sounds reasonable.

**Weaknesses:**

1. The proposed method is only applicable to the application domain where the notion of depth is meaningful. If the depth is not a crucial feature that the model utilizes, for example, in text recognition or face recognition, the proposed method may not be effective.
2. The experiment lacks sufficient variations in the dataset, model, and attack methods. This paper’s contribution primarily focuses on the adversarial training practice, so demonstrating the effectiveness of the method in various setups is an essential part of the experimental demonstration.
    * Only one dataset (ImageNet-100) was used for the experiment. The authors provided some justification about why ImageNet-100 was tested, but it also sounds like the proposed method is not applicable to the dataset that was not used in the experiments.
    * Only one model (ViT-Base) was used for the experiment. The effectiveness of the training methods should be validated over different models.
    * Only two attack methods (10-step PGD and AutoAttack Ensemble) are used. First, PGD with 10 steps does not sound like a strong enough attack. Additionally, since the 10-step PGD is already used in the adversarial training, the AutoAttack ensemble is the only attack that is new to the trained model. While this ensemble includes multiple attack methods, it should not represent various attack methods that the attacker can choose from.
3. There is only one baseline defense, which is the vanilla PGD-AT. The vanilla PGD-AT has been improved since its proposal, and more powerful adversarial training baselines have emerged. The proposed methods should be compared to one of the cutting-edge improvements for adversarial training.
4. The presented result (Table 1) is not impressive enough. Because the baseline is the vanilla PGD-AT, I doubt that the proposed method can beat other cutting-edge adversarial training baselines.
5. The robustness of the depth prediction should also be evaluated and presented in the paper. If this depth map is affected by the adversarial attack, then aligning the attention to the depth saliency (computed from the depth prediction) may not be a robust method for improving adversarial training.
6. The experiment should consider an adversary who would adapt the attack according to the proposed method. The primary hypothesis is that it should be challenging for an adversary to generate an adversarial example without significantly altering the depth map. While Section 4 discusses such hardness, some justifications should be validated through experiments.
    * One argument for the hardness is that the adversary will need to focus on some small depth-edge region. Considering that there are attacks that try to minimize the number of pixels to perturb, I don’t think that this is a strict restriction for an adversary.

**Questions:**

1. Section 4 ‘s title is “theoretical analysis”, but I don’t think that the section derives any theoretically meaningful results. “Discussions” would be a better title for the section.
2. Can the authors justify that the proposed method is still valuable for the applications where other features, e.g., the shape of some text, the texture of the surface, etc., are more dominant features than the depths?
3. Please try to add more experimental setups, including more datasets, models, and attack methods.
4. Include more baseline methods that outperform vanilla adversarial training.
5. Please provide some empirical evidence that the depth estimator used in the paper is indeed robust against adversarial perturbations.
6. Add experiments showing that an adversary who only focuses on the depth-edge region indeed fails.

---

### Note · Authors · 2025-11-12

I have read and agree with the venue's withdrawal policy on behalf of myself and my co-authors.